# Molecular Dynamics Studies of Poly(Lactic Acid) Nanoparticles and Their Interactions with Vitamin E and TLR Agonists Pam_1_CSK_4_ and Pam_3_CSK_4_

**DOI:** 10.3390/nano10112209

**Published:** 2020-11-05

**Authors:** Simon Megy, Stephanie Aguero, David Da Costa, Myriam Lamrayah, Morgane Berthet, Charlotte Primard, Bernard Verrier, Raphael Terreux

**Affiliations:** 1PRABI (Pôle Rhône-Alpes de Bioinformatique), Lyon Gerland, UMR5305, LBTI, IBCP, Lyon 1 University, 7 Passage du Vercors, 69367 Lyon CEDEX 07, France; simon.megy@ibcp.fr (S.M.); stephanie.aguero@ibcp.fr (S.A.); 2Colloidal Vectors and Tissue Transport, UMR5305, LBTI, IBCP, Lyon 1 University, 7 Passage du Vercors, 69367 Lyon CEDEX 07, France; david.da-costa@adjuvatis.com (D.D.C.); myriam.lamrayah@ibcp.fr (M.L.); berthetmorgane@gmail.com (M.B.); Bernard.verrier@ibcp.fr (B.V.); 3Adjuvatis, 7 Passage du Vercors, 69007 Lyon, France; charlotte.primard@adjuvatis.com

**Keywords:** polylactic acid, nanoparticles, molecular dynamics, dissipative particle dynamics, Pam_3_CSK_4_, Pam_1_CSK_4_, vitamin E, Toll-like receptor

## Abstract

Poly(lactic acid) (PLA) nanoparticles (NPs) are widely investigated due to their bioresorbable, biocompatible and low immunogen properties. Interestingly, many recent studies show that they can be efficiently used as drug delivery systems or as adjuvants to enhance vaccine efficacy. Our work focuses on the molecular mechanisms involved during the nanoprecipitation of PLA NPs from concentrated solutions of lactic acid polymeric chains, and their specific interactions with biologically relevant molecules. In this study, we evaluated the ability of a PLA-based nanoparticle drug carrier to vectorize either vitamin E or the Toll-like receptor (TLR) agonists Pam_1_CSK_4_ and Pam_3_CSK_4_, which are potent activators of the proinflammatory transcription factor NF-κB. We used dissipative particle dynamics (DPD) to simulate large systems mimicking the nanoprecipitation process for a complete NP. Our results evidenced that after the NP formation, Pam_1_CSK_4_ and Pam_3_CSK_4_ molecules end up located on the surface of the particle, interacting with the PLA chains via their fatty acid chains, whereas vitamin E molecules are buried deeper in the core of the particle. Our results allow for a better understanding of the molecular mechanisms responsible for the formation of the PLA NPs and their interactions with biological molecules located either on their surfaces or encapsulated within them. This work should allow for a rapid development of better biodegradable and safe vectorization systems with new drugs in the near future.

## 1. Introduction

In the past few decades, nanomaterials have emerged as a research field with a very broad range of applications in areas such as chemistry, physics, biology, medicine and pharmaceutical science [1,2,3,4,5,6]. In the particular field of drug delivery, NPs provide several advantages, including (1) improved water-solubility of poorly soluble drugs [7,8,9], (2) enhanced drug stability in physiological environments [10,11,12], (3) controlled and sustained drug release [13,14,15,16] and (4) enhanced bioavailability [17,18,19]. Such particles are commonly made of non-toxic, amphiphilic, self-assembling block polymers. In particular, the physico-chemical properties of particles made from biodegradable polymers have been extensively investigated over the last twenty years. As a result, biodegradable polymers are used in a wide array of medical applications, such as wound healing, tissue engineering, orthopedic devices, cardiovascular applications and vaccine administration [20]. In addition, the use of biodegradable polymers for NP production is safe and reliable. Moreover, particle size can be tuned by adjusting the polymer concentration and by changing the synthesis method [21]. This makes biodegradable NPs ideal vectors for drug and protein delivery, and also excellent candidates for the future of vaccine administration [22,23,24].

NPs can be produced from a wide range of biodegradable polymers, many of them having already been approved by the U.S. Food and Drug Administration (FDA). Among them, polylactic acid (PLA) has been widely investigated for drug release [25,26] and for vaccine delivery [27,28]. Indeed, the nontoxic effects of PLA and its degradation products make it a suitable candidate for biomedical application in drug delivery systems [29]. However, drug delivery remains a prominent challenge [30], mostly because of our limited understanding of all the numerous biological, chemical and physical processes at stake.

Molecular modeling can be used to provide a better understanding of how man-made NPs and various biomolecules interact together. Numerous examples have been studied in the past, including mostly gold [31,32] and carbon [33,34,35] NPs interacting with water molecules. Later on, several studies were published describing the molecular modeling of the interaction between proteins and peptides with gold NPs [36,37]. Silver NPs interacting with ubiquitin molecules have also been characterized using computational approaches [38], as have silica NPs interacting with various proteins, such as cytochrome C, RNase and lysozyme [39]. More recently, full atomistic simulations have been used to describe and predict the drug loading capacities of polymer NPs, namely, PLA-PEG NPs [40] used as nanoscale drug delivery systems.

Computational simulations provide an effective tool for gaining insights into the interaction of NPs with biomolecules Although many studies of molecular dynamics on biomolecules and nanomaterials have been conducted over the last twenty years, their application to a better understanding of how these systems interact at an atomic level remains a challenge due to the huge sizes of the molecular systems involved. In fact, it is still extremely hard to make the full connection from atomistic length-scale to the macroscopic world, even though the behavior of these networks on larger scales is now reasonably well understood.

To gain insights into this problem, alternative methods can demonstrate great efficiency. In particular, a certain mesoscopic simulation technique for hydrodynamic behavior called dissipative particle dynamics (DPD) was first introduced by Hoogerbrugge and Koelman [41,42]. This technique is based on the simulation of soft spheres, whose motion is directed by collision rules. Polymers can also be simulated with the same technique by introducing bead-and-spring type particles [43,44]. Shortly after, the theoretical foundations of DPD dynamics were established by Español and Warren [45] who studied the fluctuation–dissipation theorem in connection with this method.

In the DPD approach, several atoms are united into a single simulation particle. Thus, this mesoscopic technique opens the way to large scale simulations, with systems encompassing up to millions of atoms. This allows one to bridge the gap from atomistic simulations to mesoscopic simulations wherein mesophases and network and particle formation can be studied [46]. With this method, one first performs simulations of molecular fragments using an all-atomistic approach and force fields accounting for all the atomistic details. This allows one to derive and calculate the particular solubility parameters for all the groups of atoms previously defined. Then these results can be used as inputs to a DPD simulation to study the formation of micelles, networks, particles and so forth. One of the first successful uses of this approach was the mesoscopic simulation of cell membrane damage by Groot and Rabone [47], where the authors evidenced that the membrane structure they obtained through their DPD approach quantitatively matched with full atomistic simulations.

In this study, our work highlights how DPD can be used as a successful investigation tool to better simulate and describe the formation of PLA NPs and their interactions with various molecules of biological interest. In this paper, we focus on the biological roles of three molecules interacting with these PLA NPs during formation—first two small lipopeptides, and then the vitamin E molecule.

TLRs are a class of cell surface receptors involved in the front line of the immune response [26,48]. Lipopeptides that are N-acetylated via glycerol-linked cysteines have been shown to be active against TLRs, which led to the development of molecules such as Pam_1_CSK_4,_ Pam_2_CSK_4_ and Pam_3_CSK_4_. These commercially available molecules respectively possess one, two and three palmitoyl (Pam) hydrophobic chains. In Pam_1_CSK_4,_ this chain is directly linked to a charged hydrophilic CSKKKK peptide. In Pam_2_CSK_4_ or Pam_3_CSK_4_, the two or three hydrophobic chains are linked to a modified glycerol unit, which is linked to the hydrophilic CSKKKK peptide. Loading these amphiphilic lipopeptides-onto PLA NPs would create a biodegradable delivery system with promising immune modulating properties.

Vitamin E is also a molecule which could benefit from better vectorization systems, as it exhibits strong hydrophobic properties and has very poor water solubility. A nanoscale, biodegradable carrier such as PLA NPs could greatly improve its bioavailibity through better biodistribution and thus a better exploitation of its anti-oxidant effects.

We previously reported results for the study of the Pam_3_CSK_4_ molecule interacting with PLA NPs [26]. In this paper, we presented the successful experimental vectorization of Pam_3_CSK_4_ into PLA NPs by nanoprecipitation with a 99% entrapment efficiency and confirmed that the Pam_3_CSK_4_-loaded PLA NPs maintained their bioactivity on the TLR2 class of receptors. We also presented early results of molecular modeling for mechanisms involved in the nanoprecipitation formation process of PLA NPs, and their interactions with Pam_3_CSK_4_ molecules.

Here we present an updated version of our molecular modeling results, with a more comprehensive approach including new molecules, such as Pam_1_CSK_4_ and vitamin E, improved simulation parameters, longer simulation times, larger simulation systems and high-quality videos. We used some of our previous results to develop and validate the models presented in this study. In particular, we used our previous Pam_3_CSK_4_ calculations as a reference for further simulations of Pam_1_CSK_4_ and vitamin E. This allowed us to compare our results for the different molecules and thus validate our simulation parameters and protocols.

The two main goals of this work consisted of: (1) studying the PLA encapsulation processes of biological molecules of interest with different physicochemical properties and (2) establishing and validating a simulation protocol which could be used to predict the behavior of new molecules for future PLA NP encapsulation. In this regard, we first describe a molecular modeling comparison between the respective behaviors of Pam_1_CSK_4_ and Pam_3_CSK_4_ in concentrated solutions. Then we go through the simulations of the nanoprecipitation processes for the formation of PLA NPs interacting with these two amphiphilic lipopeptides separately. Last, as a valid comparison point, we show similar computer simulations on the more hydrophobic vitamin E molecule. We compare our modeling results with the experimental in vitro results for the encapsulation of vitamin E also described in this paper. Finally, we compare the encapsulation processes of the three molecules reviewed in this article—vitamin E, Pam_1_CSK_4_ and Pam_3_CSK_4_.

## 2. Materials and Methods

### 2.1. Pam_3_CSK_4_ Encapsulation into PLA NPs

Pam_3_CSK_4_‘s in vitro encapsulation process has already been described [26]. Protocols for in vitro preparation of PLA NPs and for the Pam_3_CSK_4_ entrapment remained unchanged.

### 2.2. Vitamin E Encapsulation into PLA NPs

Polymeric NPs were produced by nanoprecipitation adapted from Fessi et al. [49]. PLA polymer was produced in the laboratory following the patent FR2745005A1 of Phusis (Grenoble, France) [50]. In summary, 1 g of poly(d,l-lactic acid) (MW = 50 kDa) was dissolved in 49.25 mL of acetone (Carlo Erba, Val de Reuil, France). Vitamin E ((±)-α-Tocopherol, Sigma-Aldrich, Saint-Quentin Fallavier, France) was dissolved in acetone and 750 µL was added into the PLA solution. This organic phase was poured by fast drop-by-drop into an aqueous phase containing 20 mL of water (Aguettant, Lyon, France) and 15 mL of ethanol (Carlo Erba, France) under mild stirring. Acetone and ethanol were evaporated under low pressure at 32 °C using a rotavapor (Buchi, Rungis, France).

### 2.3. NP Physicochemical Characterization

Dynamic Light Scattering (DLS) experiments were performed with a Zeta Sizer Nano ZSP (Malvern Panalytical, Royston, UK) to evaluate the hydrodynamic diameters of plain and vitamin E-loaded NPs and their polydispersity indexes (PdI) after their dilution in a 0.22 µm-filtered 1 mM NaCl solution (Sigma Aldrich, France). The zeta potentials of the particles were estimated by electrophoretic mobility using the same apparatus.

### 2.4. Quantification of Vitamin E Loading

Vitamin E-loaded NPs were centrifuged for 30 min at 16,896× *g* and supernatants were collected to determine the quantity of non-encapsulated vitamin E. Pellets containing the encapsulated drug were washed twice with water (Aguettant, Lyon, France) and resuspended in HPLC-grade acetonitrile (Carlo Erba, France). Vitamin E was quantified by HPLC (1100 Series, Agilent, CA, USA) with a silica-based reversed-phase column (C18, 5 μm, Merck, Lyon, France) by injecting 40 µL of sample at 25 °C. An isocratic method was performed with a mobile phase containing acetonitrile, water and trifluoroacetic acid (Merck, France) in the proportions 99:1:0.1 (*v*/*v*/*v*) at 0.8 mL/min. The detection wavelength was fixed at 290 nm and a linear calibration curve was obtained from 1 to 80 µg/mL (*R*^2^ = 0.9988). The loading rate and the encapsulation efficiency were calculated using the following equations:$$\mathrm{Loading}\;\mathrm{rate}=100\times \frac{\mathrm{mass}\;\mathrm{of}\;\mathrm{encapsulated}\;\mathrm{vitamin}\;E}{\mathrm{mass}\;\mathrm{of}\;\mathrm{PLA}}$$
$$\mathrm{Encapsulation}\;\mathrm{efficiency}=100\times \frac{\mathrm{mass}\;\mathrm{of}\;\mathrm{encapsulated}\;\mathrm{vitamin}\;E}{\mathrm{total}\;\mathrm{mass}\;\mathrm{of}\;\mathrm{vitamin}\;E}$$

### 2.5. Molecular Dynamics

DPD simulations were performed using the Materials Studio software (Materials Studio 2017, Biovia, Dassault Systèmes, Lyon, France, 2016). For the water molecules (W), a coarse graining approach was used, in which one bead represents 3 molecules of water. The radius of the water bead was set to 3.23 Å and its molecular mass to 54 Da. Pam_1_CSK_4_ molecules were constructed using 5 different types of beads: lysine (K), serine (S), modified cysteine (C), modified glycerol (G) and five C_3_H_6_ fatty acid units (F). Pam_3_CSK_4_ molecules were constructed using the same C, S, K and F beads. PLA molecules were constructed as linear repetitions of 70 units of lactic acid monomers (LA)*_n_*. Vitamin E molecules were constructed using 2 different types of beads: one for the chromanol double cycle (CHR), and 3 beads (each composed of 5 carbon atoms) for the fatty acid chain (C5). All the different kinds of beads used for our calculations are summarized in Figure 1.

For the calculations, all the solubility parameters *δ_i_* were calculated with the Materials Studio software using either the Synthia module for the beads presented as polymers (LA, F, C5), or models constructed with the Amorphous Cell module for the beads presented as monomers in the simulation. For the models constructed as amorphous cells, all the calculations were performed with the Forcite module using the COMPASS II [51,52] force field for atom parameters and partial charges. After a preliminary geometry optimization phase, a preliminary short equilibration run of dynamics with NVT parameters was performed (500 steps), followed by a longer production run with NVE parameters (5000 steps). The corresponding solubility parameters were then used to calculate the Flory–Huggins interaction parameters *χ_ij_* for the corresponding binary mixtures, which were calculated using the relation *χ_ij_* = (*v*/*RT*)(*δ_i_* − *δ_j_*)^2^ where *R* is the gas constant, *T* the absolute temperature and *v* is the mean volume per mole of the two corresponding components [47]. The molar volume for each component was determined using the Molecular Operating Environment software (MOE 2017, Chemical Computing Group, Köln, Germany). The Flory–Huggins interaction parameters were then converted into DPD repulsion parameters *a_ij_*, which were obtained using the relation *a_ij_* = 25 + 3.50 ×*χ_ij_* [46]. All the DPD repulsion parameters used are summarized in Table 1 and Table 2.

For the self-assembly studies of Pam_3_CSK_4_ and Pam_1_CSK_4,_ we used 280 × 280 × 280 Å cubic periodic boxes of water containing randomly distributed molecules of Pam_3_CSK_4_ or Pam_1_CSK_4_. We used 3%, 10% and 30% concentrations (number of Pam_3_CSK_4_ or Pam_1_CSK_4_ molecules versus vs. water molecules). After an initial geometry optimization within the Mesocite module, DPD experiments were run using a total simulation time of 10 ns.

For the particle formation, our calculations were performed using 280 × 280 × 280 Å cubic periodic boxes of water as starting points for all the dynamics. Droplets with a radius of 130 Å containing a mixture of water; PLA; and Pam_3_CSK_4_, Pam_1_CSK_4_ or vitamin E molecules were placed at their center. In order to mimic the experimental ratio of PLA vs. Pam_3_CSK_4_ or Pam_1_CSK_4_ molecules during the NP synthesis, we used inside of the 130 Å droplet a mixture ratio of 201 water molecules vs. 67 PLA molecules vs. 1 Pam_x_CSK_4_ molecule, thereby mimicking a Pam_x_CSK_4_ concentration of 180 μg/mL and a 4% PLA/water (w/w) ratio. For the vitamin E simulations, we used a ratio of 750 water molecules to 125 PLA chains and 117 vitamin E molecules in order to recreate an 8% w/w ratio for vitamin E vs. the PLA chains. After an initial geometry optimization within the Mesocite module, DPD experiments were conducted using a time scale of 3 ps and various total simulation times, up to 30 ns.

For the Pam_3_CSK_4_/PLA saturation studies, we used 500 × 100 × 100 Å periodic boxes of water. A layer of PLA chains (with a thickness of 100 Å) was placed at one extremity, and the rest of the box was filled with increasing concentrations of randomly dispersed Pam_3_CSK_4_ molecules solvated in water. We show our results with three molecular ratios of Pam_3_CSK_4_ vs. water (number of molecules vs. number of molecules), ranging from 1% to 10%. After an initial geometric optimization within the Mesocite module, DPD experiments were conducted using a time scale of 3 ps and longer simulation times, up to 300 ns.

## 3. Results

### 3.1. Behavior of Pam_1_CSK_4_ and Pam_3_CSK_4_ at Increasing Concentrations

In order to validate our approach and the used simulation parameters, we confronted the molecular behavior of Pam_3_CSK_4_ and Pam_1_CSK_4_ in a series of experiments, using each molecule separately in an aqueous environment at increasing concentrations, without any PLA molecules. Thus, we performed DPD simulations using the Materials Studio software. This non-atomistic kind of simulation is particularly well-adapted to large systems. We started all of these simulations in 280 Å periodic cubic boxes with randomly distributed molecules mixed with water. After an initial geometric optimization, we started the DPD experiments for a total simulation time of 10 ns, which is a sufficient time to observe the formation of micelles [47], and in this case allows one to reach a stable state.

As shown in Figure 2, at a 3% concentration (number of Pam_3_CSK_4_ or Pam_1_CSK_4_ molecules vs. number of water molecules), little difference was observed between the two molecules, which tended to auto-assemble and form small micelles surrounded by large clusters of water molecules. Those micelles are characterized by concentrations of the hydrophobic fatty acid chains (in green) at their centers, whereas the more hydrophilic peptidic parts (K residues, in purple) were directly exposed to the water on their outer surfaces. At 10% concentration, Pam_3_CSK_4_ molecules started to form bigger clusters with more complex internal self-assembly motifs, which look like flattened micelles of some kind, whereas Pam_1_CSK_4_ molecules remained in regular organized micelles, with the same hydrophilic/hydrophobic separation of their constitutive parts. Finally, at a 30% concentration, only Pam_3_CSK_4_ molecules adopted a different type of well-defined multimeric structure, creating extended nanotubes of some kind, along with some more regular spherical micelles, whereas Pam_1_CSK_4_ molecules remained in regular (though bigger) micelles, with a diameter of around 5 nm.

These structures are extremely reminiscent of those observed in two recent papers. First, the self-assembly of the Pam lipopeptides was examined using a combination of electron microscopy and small-angle X-Ray scattering (SAXS) techniques [48]. Using cryo-transmission electron microscopy (TEM) images, it was shown that Pam_1_CSK_4_ (and also Pam_2_CSK_4_) molecules form spherical micelles in concentrated solutions, with a measured diameter of 5 nm, which is in total agreement with our simulations. Furthermore, in a complete contrast, Pam_3_CSK_4_ molecules form flexible “wormlike micelles coexisting with regular micelles,” which is precisely what was observed at the end of our simulations. The SAXS data indicate a 5.3 nm thickness for the Pam3CSK_4_ bilayers in the wormlike structures, which is excellent agreement with our simulations as well. The cryo-TEM pictures are reproduced in Figure 1 with the agreement of their author.

Moreover, those “flexible flattened wormlike micelles” were also observed using all atomistic molecular dynamic simulations [53] with a smaller (45 to 180) number of Pam_3_CSK_4_ molecules. The resulting calculated structures are in excellent agreement with our models. This further validates our non-atomistic DPD approach, which gives similar results with shorter computing times, but also allows for the study of larger systems, with more lipopeptides molecules, and with or without the addition of numerous PLA chains.

### 3.2. Pam_3_CSK_4_ and Pam_1_CSK_4_ Particle Formation

Different techniques are available for synthetizing PLA-based nano and micro particles: single, double and multiple emulsion methods, precipitation-based methods, direct compositing methods, microfluidic technique and the hydrogel template method [29]. The nanoprecipitation method used in our studies allows for a narrow size distribution and requires a low energy source. Moreover, in contrast to the emulsion-based methods which require the use of surfactants, no additional molecule is added during the nanoprecipitation process described in this study: the molecules of interest are directly entrapped thanks to the hydrophobic properties of the PLA chains, and the water molecules are quickly excluded from the particle core. Less toxic solvents are used, and there are no residual solvent traces left at the end of the particle formation. Thus, the molecular modeling simulation of such systems does not require the addition of surfactant molecules either.

In order to gain insights into the nanoprecipitation-based particle formation process, we also performed DPD simulations for biologically relevant molecules in interaction with PLA chains. Our DPD simulations allow one to model large systems, in a range of sizes from nm to μm, and are particularly well-adapted to PLA-NP studies. In order to mimic the particle formation process, we used cubic periodic boxes of water measuring 280 × 280 × 280 Å as starting points for all the dynamics. Droplets with a radius of 130 Å containing mixtures of water; PLA; and Pam_3_CSK_4_ or Pam_1_CSK_4_ were placed at their center. The experimental particle formation of PLA NPs with Pam_3_CSK_4_ has already been previously described. We used a PLA/Pam molecular ratio of 67/1, which perfectly mimics a PLA/water 4% mass ratio and a Pam molecular concentration of 180 μg/mL. Those are the exact conditions which were used during our laboratory assays for the nanoprecipitation of the particles. After an initial geometry optimization, we started the DPD dynamics and allowed the simulations to run for times up to 30 ns.

For Pam_1_CSK_4_, the results are essentially the same as those already described for Pam_3_CSK_4_. In both cases, at the very start of the simulations, we observed that the PLA polymer chains underwent a volume contraction and released all the Pam_3_CSK_4_ molecules outside of the PLA particle in formation. Starting from a 260 Å diameter, the final diameter measured for the complete NP was about 150 Å in each case. Most of the water molecules were rapidly excluded as well. As a result, no Pam_3_CSK_4_ or Pam_1_CSK_4_ molecules were found deeply buried inside the PLA particle, as illustrated in Figure 3, and in Appendix A. The Pam molecules interacted with the PLA chains by their fatty acid chains, while keeping their more hydrophilic parts (mostly the Lys residues) pointing outwards from the particle, directly exposed to the surrounding water. In both cases, the Pam molecules were randomly distributed on the surface and interacted with the PLA chains through their buried fatty acid chains.

Interestingly, during the simulations, some PLA chains form small clusters, which are independent of the main particle and may or may not encompass some Pam molecules. Some of them are gradually absorbed by the main particle along the way, whereas some remain isolated until the end of the experiments, as shown in Figure 3D. This suggests that the number of PLA chains inside a particle may be somehow limited to a certain number, which is consistent with the fact that the PLA chains form multiple NPs of regular size during the laboratory assays. This also goes along well with the increase of the PdI previously observed [26] upon the formation of Pam_3_CSK_4_ loaded PLA NPs. In contrast, no Pam_1_CSK_4_ or Pam_3_CSK_4_ molecule was found isolated in solution or interacting with PLA chains at the end of each simulation. This is consistent with our Pam_3_CSK_4_ experimental encapsulation results where we previously demonstrated a 99% entrapment efficiency [26].

### 3.3. Pam_3_CSK_4_/PLA Saturation Studies

Despite providing satisfactory models for the Pam/PLA particle formation process, our results show only a small number of lipopeptides molecules being incorporated into the PLA NPs. This is mainly due to the size of our simulated systems, which remained smaller than the real-life particles observed in our laboratory. However, as it seemed that we were very far from the saturation of the PLA NPs with lipopeptides molecules, we tried to increase their number and see how many our simulations would allow us to incorporate within a PLA layer of a given size.

In this regard, we used 500 × 100 × 100 Å periodic boxes of water filled with a 100 Å PLA layer at one extremity, and with randomly dispersed molecules of Pam_3_CSK_4_ at different concentrations, as illustrated in Figure 4. It should be noted that the PLA layer is thick enough so that molecules of Pam_3_CSK_4_ located on each side of the surface cannot interact together through their buried fatty acid chains. After an initial geometry optimization, we performed DPD experiments in order to observe the diffusion of the lipopeptides molecules and their incorporation into the PLA layer. We used longer simulation times (up to 300 ns) in order to allow for homogenous repartition of the molecules inside the simulation box.

Results are presented with three different concentrations of lipopeptides (number of Pam3CSK4 molecules vs. number of water molecules), 1%, 5% and 10%. Interestingly, at 1%, all the Pam_3_CSK_4_ molecules end up located at the surface of the PLA layer. A comparison with the previous results of Figure 3 confirms that a higher density of Pam_3_CSK_4_ molecules could be well obtained. At the concentration of 5%, more lipopeptides molecules are located at the surface of the PLA layer, but some of them start to form small regular spherical micelles similar to the ones observed in Figure 2. At this point, a close analysis reveals that 75% of the Pam_3_CSK_4_ molecules are located inside the PLA layer, whereas 25% form micelles away from the PLA layer.

At the 10% concentration, the number of Pam_3_CSK_4_ molecules located at the surface of the PLA layer is even greater than in the previous 5% experiment. However, as the Pam_3_CSK_4_ saturation of the PLA layer increases, it should be noted that the proportion of Pam_3_CSK_4_ molecules located within the PLA layer decreases in regard to the total number of Pam_3_CSK_4_ molecules involved in the simulation. We can see that only 60% of the total Pam_3_CSK_4_ molecules of the simulation were located inside the PLA layer, whereas 40% were present in the surrounding micelles.

These results evidence that there is some kind of competition between the PLA layer and the formation of micelles for the incorporation of Pam_3_CSK_4_ molecules. At first the PLA layer seems to demonstrate a better affinity for the lipopeptide molecules. Then, with the gradual saturation of the PLA chains, the residual molecules tend to start forming micelles. This observation is in clear agreement with our previous laboratory assays, where we demonstrated that the vectorization of lipopeptides such as Pam_3_CSK_4_ was possible even at concentrations above their critical aggregation concentration (CAC) [26].

### 3.4. Vitamin E Encapsulation

To balance the studies of lipopeptides presented in this article, we wanted to extend our simulation models to other molecules with radically different properties. The main characteristic of lipopeptides accounting for their behavior with respect to PLA chains is their amphiphilic property, which allows them to be located at the surfaces of PLA NPs, with their hydrophobic lipidic extremities buried inside the particle and their hydrophilic peptidic extremities pointing towards the surrounding water. To challenge our simulation parameters and further validate our DPD approach, we started the study of a highly hydrophobic molecule—vitamin E. This was meant to allow us to develop and validate our DPD modeling approach presented in this study, by comparing the results for molecules with drastically different properties, and thus validate our simulation parameters and protocols. In this regard, we studied the entrapment of vitamin E molecules within PLA NPs, in both in vitro and DPD simulations, and then confronted the laboratory assays with the simulation results.

PLA NPs were produced by nanoprecipitation using increasing concentrations of vitamin E, followed by an evaluation of their colloidal characteristics and the encapsulation efficiency. The hydrodynamic diameter of plain NPs measured by DLS was equal to 190 nm and their PdI of 0.05 pointed out a narrow size distribution (Figure 5). We observed that the colloidal properties of the PLA NPs were impacted upon their loading with vitamin E. The hydrodynamic diameter increased up to 220 nm and their size distribution was more heterogeneous with a PdI reaching 0.12. The encapsulation efficiency, evaluated by HPLC, was around 85% and remained unchanged at loading rates ranging from 1.2% to 10%. Both the plain and the vitamin E-loaded NPs conserved a zeta potential estimated by electrophoretic mobility close to −60 mV, and the suspensions were stable for at least two months at 4 and 37 °C (data not shown).

To better understand the experimental results obtained with vitamin E, we performed DPD experiments in order to evaluate its ability to be encapsulated within PLA NPs. As vitamin E is a much more hydrophobic molecule than the previously used lipopeptides, we expected it to be fully buried inside the PLA NPs. We used several PLA to vitamin E weight to weight ratios (loading rates) starting from 0.5% and going up to 8% in order to explore different options. The results shown here were obtained with an 8% *w*/*w* ratio for vitamin E vs. the PLA chains, which is close to the maximum tested during the in vitro experiments. After an initial geometry optimization, DPD experiments were run using a total simulation time of 30 ns. The results of our DPD experiments are shown in Figure 6 and illustrated in Appendix A.

During the fast particle formation, all the water molecules were excluded from the hydrophobic core of the particle. We also observed a volume contraction, from a 260 Å diameter at the beginning to about 140 Å after 30 ns of simulation time. During the particle formation simulations, some PLA chains along with some vitamin E molecules assembled and formed small independent clusters. Most of them were absorbed by the central particle during the few first ns of the DPD calculations (as illustrated in Appendix A), but some clusters of isolated vitamin E remained either associated with a few PLA chains, or stayed by themselves. Then the simulation reached a stable state, with some small molecular clusters remaining isolated even after 30 ns of simulation.

These results go well along with our laboratory experiments, where the vitamin E encapsulation efficiency was demonstrated to be about 85% at similar loading rates (Figure 5b), meaning that about 15% of the vitamin E molecules are not encapsulated into PLA NPs and are free to form some small independent clusters. Interestingly, this contrasts with what was observed with the Pam lipopeptides in Figure 3, where the experimental entrapment efficiency reached 99% and no isolated Pam_1_CSK_4_ or Pam_3_CSK_4_ molecules were observed at the ends of the molecular simulations. Either way, in the case of a total entrapment efficiency or not, our modeled systems account for this experimental parameter.

The observation of these little clusters of vitamin E or vitamin E and PLA chains also agrees with the experimental observation of the PdI of the particles, which increases upon vitamin E loading (Figure 5a). This also suggests that there is a form of competition between the PLA chains and the isolated vitamin E and PLA clusters to incorporate the remaining supplemental vitamin E molecules.

The stability of the simulation also goes well along with our experimental observation that the vitamin E-loaded particles remained stable for at least two months at 4 and 37 °C. Finally, the diameter size increase of the particles (Figure 5a) upon vitamin E loading was observed in our simulations as well, where the added molecules of vitamin E contributed to forming bigger NPs, as compared to simulations with PLA chains only.

## 4. Discussion

In this article, we provided some mechanistic explanations for the rearrangement of the ligands during the nanoprecipitation process. We based our study on two sets of molecules exhibiting very different physicochemical profiles: vitamin E, and TLR agonists Pam_1_CSK4 and Pam_3_CSK4. We used DPD simulations in conjunction with biochemical experiments in order to study the nanoprecipitation process by proving that, although Pam_1_CSK4 and Pam_3_CSK4 molecules tend to respectively form either micelles or wormlike particles in concentrated solutions, they both end up located on the surfaces of the PLA NPs upon their formation. On the other hand, vitamin E molecules are buried deeper at the core of the particle when mixed with PLA due to their hydrophobic properties.

These results, which are extremely consistent both with the previous studies describing the NP formation and with our experimental studies, tend to underline the two following aspects: (1) The NP/molecule assembly depends on their respective concentrations and physicochemical profiles. This can provide us with a better understanding of the molecular mechanisms at stake and help us to predict the encapsulation profiles of other molecules based on their physicochemical properties. (2) The DPD approach that we chose in order to model the nanoprecipitation process proved to successfully describe the molecular behavior of self-assembling lipopeptides in micelles or in flattened wormlike micelles, but also the behavior of self-assembling PLA chains interacting with hydrophobic and amphiphilic molecules. Our results also indicate that DPD can be used for systems encompassing different bead sizes. Indeed, whilst DPD theory suggests that you could have beads with different masses in your simulated system, there has been so far little validation on the effect of this. This was achieved in our case without affecting the quality of the prediction, with the molecular weight of the different beads ranging from 42 Da for the F bead of Pam_1_CSK4 and Pam_3_CSK4 to 219 Da for the CHR bead of vitamin E, as illustrated in Figure 1. Moreover, the DPD method, which was originally developed mostly for polymers, gave satisfactory results here, even with small molecules such as vitamin E simulated with only two different beads, and only four in total (one CHR and three C5 beads), whereas our PLA chains were 70 LA monomers long.

However, despite the very good agreement between our experimental observations and our simulations, some points related to our modeling strategy have to be addressed. In particular, as in most molecular dynamics methods, we rather underestimated the number of water molecules in our simulations, resulting in a much more concentrated medium than the reality. We modified these parameters, because strict compliance in our simulations with the concentrations involved in vitro was nearly impossible to achieve using a realistic computing power. However, even if the number of water molecules was greatly underestimated, our ratios between LA monomers and lipopeptides molecules or between LA monomers and vitamin E molecules exactly match the reality of the in vitro experiments. Besides, reproducing the exact concentration would not provide Appendix A since the interactions between the NPs and the substrates mainly depend on short distance interactions. Adding more water molecules would just dilute the relevant molecules but would not change the nature of the molecular interactions responsible for the micelle or the particle formation. Thus, by reducing the number of modeled water molecules, we did not change the dynamics of the interactions, but we optimized our calculations in terms of power. However, it also should be noted that limiting the number of water molecules, and thus working in a more concentrated medium, enhances the attraction/repulsion behavior of the different molecules present in the system. This leads to quite a segregationist model where the hydrophobic/hydrophilic interactions are somehow overestimated. This might enhance the encapsulation behavior, which happens extremely fast in our simulations. Additionally, this simplified model is not really suited to describe long and complex processes such as the release of molecules from the PLA NPs. DPD simulations can typically describe molecular evolutions in the timescale of μs or at best ms (such as the encapsulation process), but release is a totally different process which occurs within hours or days. This is another inherent limitation of our method. In this regard, all atomistic simulations of the diffusion of an encapsulated molecule within a PLA NP could provide useful insights into the release process.

The size of our simulations is another point to be discussed. For the formation of the micelles and the wormlike micelles (280 Å periodic cubic boxes), this size is sufficient to observe the formation of self-assembling nanostructures exhibiting the exact same size as experimentally observed in the cryo-TEM images, namely, 5 nm in diameter for the of Pam_1_CSK4 micelles and slightly more for the thickness of the Pam_3_CSK4 bilayers found in the wormlike micelles.

On the other hand, in order to limit the computing power needed, our simulated PLA NPs were roughly scaled to 1/10 relatively to the real size of the PLA NPs produced and analyzed in our laboratory. In an attempt at consistency, as explained in the Materials and Methods section, we also used PLA chains about 10 times shorter: those were 70-mer chains in all our simulations, whereas the real length of the polymeric chains we used in the laboratory was more like 700 units of monomeric lactic acid. That did not much effect the outcome of the vitamin E particle formation experiments: one can expect to observe particles 10 times bigger, still with PLA chains mixed to an equivalent amount of vitamin E. The situation is a bit more complex with molecules such as Pam_1_CSK_4_ and Pam_3_CSK_4_, as these molecules are exclusively located at the surface of the particle at the end of every simulation. A 10 times bigger particle in radius (or diameter) would have a volume 1000 times bigger, but a surface only 100 times as large. By keeping the same PLA/Pam molecules ratio, this would mean that the simulation should account for 1000 times more PLA monomers, 1000 times more Pam molecules, but only a peripheral surface 100 times larger, which would increase the surface concentration of our particles. Our particle formation experiments (Figure 3) actually show that the PLA surface is far from completely saturated by Pam_3_CSK_4_ molecules. The actual formation process of a 10 times larger particle would obviously lead to a much higher density of Pam_3_CSK_4_ molecules at the surface of the particle. Our simulated models, however, are in good agreement with this observation, as this was later investigated with the saturation experiments (Figure 4). This figure highlights the fact that our simulated particles could accommodate for much more Pam_1_CSK_4_ or Pam_3_CSK_4_ molecules on their surfaces than illustrated in Figure 3.

Another discussion topic involves the duration of our simulations. Even though the formation times of our Pam_1_CSK_4_ micelles are in agreement with the DPD observations of Groot and Rabone for similar systems [47], the timescale we observed for a complete NP formation (a few ns) may be considered extremely short. The smaller size of our particles may provide some kind of explanation, but really, it is important to mention that in DPD simulations, the characteristic times and lengths of the solvent molecules are inferior to the ones of the polymer molecules by several orders of magnitude. In this regard, DPD simulations rely more on hydrodynamic considerations than Newtonian molecular dynamics. Thus, classical molecular dynamics would require time steps derived from the solvent properties, and therefore would involve extremely long simulation times to observe some major conformational changes in the organization of the polymer molecules.

The goal behind these strictly controlled adjustments is to optimize the simulation parameters and to observe the PLA NP encapsulation process in a reasonable amount of computing time. As a result, it is important to notice that timescales of a DPD simulation are not transposable to the ones observed in the context of an experimental approach, or even in a classical Newtonian molecular dynamics experiment. On a side note, it is important to mention that none of the calculations presented in this study exceeded 50 h in total simulation time, running on 40 out of an 80 processors computer, which highlights the computing efficiency of the method.

Finally, although Meunier et al. considered DPD methods less efficient than fully atomistic simulations for the estimation of the loading capacity of substrates on PLA-PEG NPs [40], we propose here a more nuanced point of view: DPD may not be as accurate as all atomistic simulations in a number of applications. However, in the context of this study, DPD perfectly matches all atomistic simulations [53] and experimental results obtained in our laboratory [26] or elsewhere [48]. Thus, we conclude that DPD, despite its inherent simplifications, represents a fast and practical method for the molecular simulation of large systems over long periods of simulation time, with a great computing efficiency. In particular, DPD can be used as a powerful simulation method to simulate the nanoprecipitation process involved in the PLA NP formation, and can help us to predict the encapsulation profile of other molecules based on their physicochemical properties.

## 5. Conclusions

In this study, we investigated the behavior of different molecules interacting with self-assembling PLA NPs, using both biochemistry approaches and molecular modeling tools. We confronted our calculations with the experimental data obtained in our laboratory, but we also took into account experimental results from other laboratories, namely, the cryo-TEM pictures provided with the consent of their authors. Our simulation results are in excellent agreement with all these experimental observations. In this regard, our work demonstrates that DPD mesoscopic simulations allow for the accurate and efficient modeling of complex systems such as NPs interacting with small molecules of biological interest.

In particular, we demonstrated that our simulated systems were able to account for the following experimental facts: (1) Pam_1_CSK_4_ and Pam_3_CSK_4_ molecules in concentrated solutions and in absence of any other molecules adopt the same behavior as what is observed in cryo-TEM experiments: Pam_1_CSK_4_ molecules form spherical micelles with a measured diameter of 5 nm, whereas in contrast, Pam_3_CSK_4_ molecules form flexible “flattened wormlike micelles coexisting with regular micelles”. (2) PLA chains auto-assemble into hydrophobic NPs in an independent process during which the vast majority of the water molecules present at the beginning of the simulation are quickly excluded from the particle core in formation. (3) Our simulated systems give very different results for molecules with very different biochemical properties: Even though both hydrophobic and amphiphilic molecules associate with the newly formed PLA NPs, they interact in a very different manner. Hydrophobic molecules such as vitamin E are buried deeply at the cores of the particles, completely mixed and intertwined with the PLA chains, whereas amphiphilic molecules such as the Pam lipopeptides only interact with the PLA chains through their hydrophobic moiety, and keep their hydrophilic constituents at the surface of the particle, free to interact with the surrounding water molecules.

This work highlights the added value of a PLA NP-based delivery system for both amphiphilic and hydrophobic molecules, such as the Pam lipopeptides and the vitamin E molecule. A very good control could be added with the study of a completely hydrophilic molecule or small chain macromolecule to validate the extreme ends of this case study—both amphiphilic to hydrophobic. Some experimental work in this direction has already been performed in our laboratory. As expected, we observed an extremely limited encapsulation efficiency and most of the hydrophilic molecules remain in the surrounding water. Of course, in this case, we highly suspect that the few encapsulated molecules end up located at the surface of the PLA NPs and are not buried inside. A DPD study of the interaction of PLA NPs with such molecules would be a very good follow up to our work and will probably be started in the near future.

Pam delivery nano-systems could very well be used for immunotherapies in a near future, whereas systems such as PLA NP-based vitamin E carriers are promising models for new drug and antibiotic delivery systems. Furthermore, PLA polymer offers the advantage of being less toxic than inorganic polymers. Our results indicate that this method should be applicable to similar amphiphilic molecules and more generally to hydrophobic molecules. This approach could also be used for vaccination purposes, as the encapsulation of immune modulators [20] could facilitate the antigen presenting cell stimulation by protecting the molecule of interest and thus potentializing its activity.

As a result, our mesoscopic DPD experiments can help to bring some mechanistic explanations to the formation process of the PLA NPs and to their interactions with a variety of biological molecules. In particular, this allows us to better describe the behavior of these molecules during the nanoprecipitation process, and then to better understand their biological properties. Our work illustrates that DPD can be used as a molecular modeling tool of choice for further applications in various fields, such as drug delivery, immunotherapy, vaccines and cancer treatment. Our results can be used to better understand the nature of drug–polymer interactions, which could lead to the development of better delivery systems in the near future.

## Figures and Tables

**Figure 1 nanomaterials-10-02209-f001:**
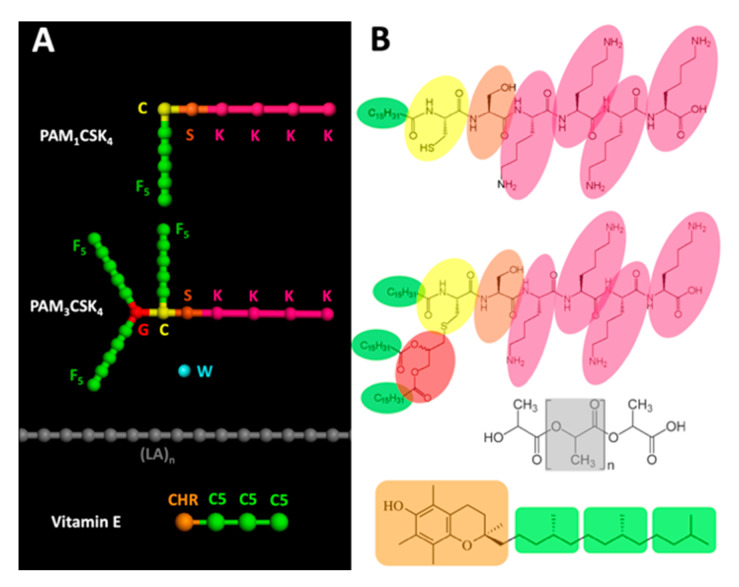
(**A**) Schematic representation of the 9 different kind of beads used for the dissipative particle dynamics (DPD) calculations. Lysine (K), serine (S), modified cysteine (C), modified glycerol (G) and five C_3_H_6_ fatty acid units (F) for the Pam_1_CSK_4_ and the Pam_3_CSK4 molecules; water (W), where one bead represents 3 molecules of water; linear chains of lactic acid monomers (LA)*_n_* for the PLA chains; chromanol (CHR) for the double cycle; and 3 beads (C5) each composed of 5 carbon atoms for the fatty acid chain. (**B**) Representations of the Pam_1_CSK_4_, Pam_3_CSK4, water, PLA and vitamin E molecules. The groups of atoms corresponding to the different beads are evidenced using colored backgrounds.

**Figure 2 nanomaterials-10-02209-f002:**
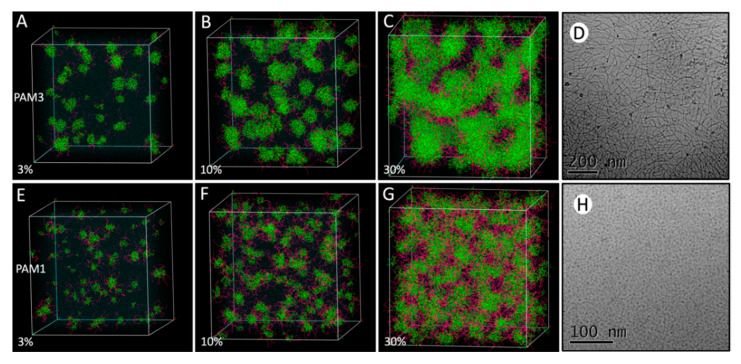
Comparison of DPD simulations for Pam_3_CSK_4_ and Pam_1_CSK_4_ in 280 Å simulation boxes: (**A**–**C**) Pam_3_CSK_4_ DPD simulation and (**E**–**G**) Pam_1_CSK_4_ DPD simulation. The relative number of Pam molecules vs. the number of water molecules is indicated as follows: (**A**,**E**) 3%, (**B**,**F**) 10%, (**C**,**G**) 30%. PLA and Pam molecules were randomly mixed in the full simulation box. The total DPD simulation time was 10 ns in every case. The results are displayed for each simulation experiment. (**D**,**H**) Cryo-TEM images of the self-assembled structures, courtesy of Ian W. Hamley [48].

**Figure 3 nanomaterials-10-02209-f003:**
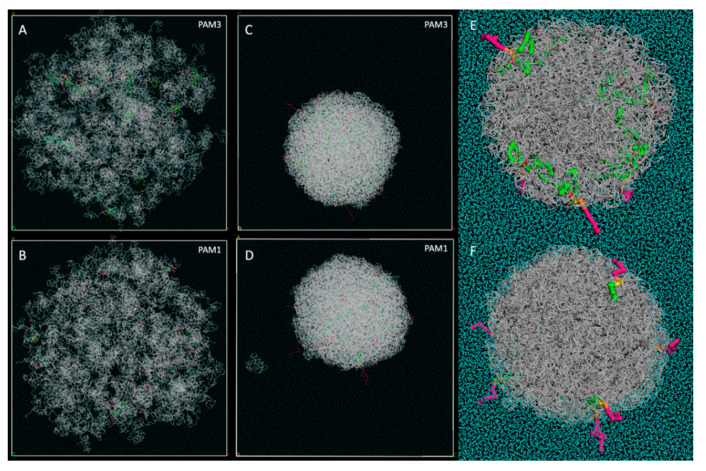
Comparison of DPD simulations for the PLA particle formation with Pam_3_CSK_4_ and Pam_1_CSK_4_: (**A**,**B**) Starting point of the DPD simulation: (**A**) Pam_3_CSK_4_, (**B**) Pam_1_CSK_4_. Droplets with a radius of 130 Å containing a mixture of water, PLA and Pam molecules were placed at the center of a water-filled simulation box. (**C**,**D**) End of the simulation after 30 ns total simulation time: (**C**) Pam_3_CSK_4_, (**D**) Pam_1_CSK_4_. Pam_3_CSK4 and Pam_1_CSK4 hydrophilic parts (in magenta) were excluded from the PLA nanoparticle, and the molecules interacted with the PLA surface through their buried fatty acid chains (in green). The scale factor is the same as in (**A**,**B**) for better observation of the volume contraction. The whole simulation cell (280x280x280 Å) is displayed in A, B, C and D. (**E**,**F**) Zoomed-in cross-section of the particle. The buried fatty acid chains (in green) of either Pam_3_CSK_4_ (**E**) or Pam_1_CSK_4_ (**F**) are clearly visible inside the spherical core of the particle.

**Figure 4 nanomaterials-10-02209-f004:**
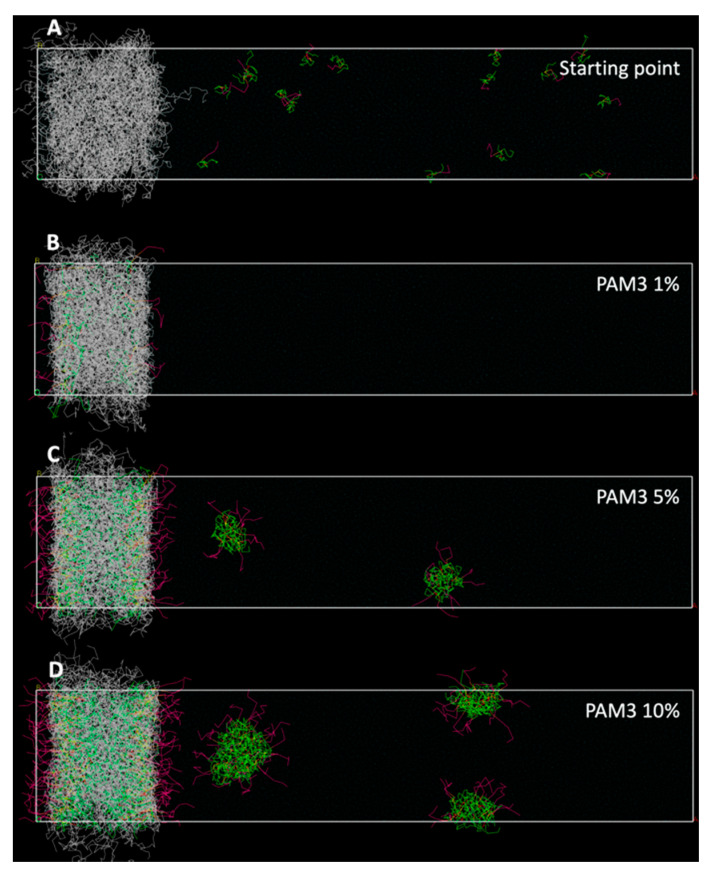
Comparison of DPD simulations for the Pam_3_CSK_4_/PLA saturation studies: (**A**) Starting point of the simulations: the PLA layer is still relaxed and the Pam_3_CSK_4_ molecules are randomly distributed inside the simulation box. (**B**–**D**) End of the simulations after 300 ns of simulation, with increasing concentrations of Pam_3_CSK_4_ molecules.

**Figure 5 nanomaterials-10-02209-f005:**
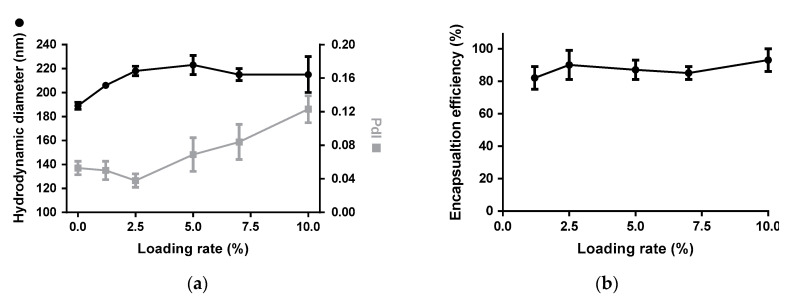
(**a**) Physicochemical characteristics (hydrodynamic diameter and PdI) and (**b**) encapsulation efficiency of vitamin E-loaded PLA NPs for increasing loading rates. Values are means ± standard deviations of four measurements for one representative experiment.

**Figure 6 nanomaterials-10-02209-f006:**
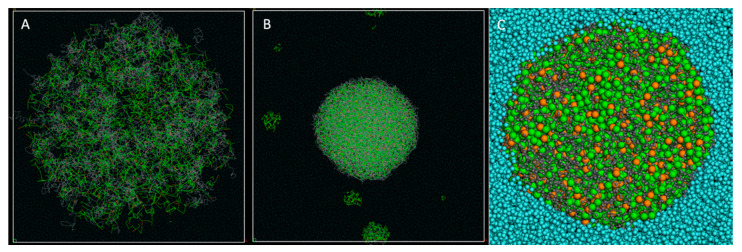
DPD simulation for the PLA particle formation with vitamin E: (**A**) Starting point of the DPD simulation. A droplet with a radius of 130 Å containing a mixture of water, PLA and vitamin E molecules (8% loading rate) is placed at the center of a water-filled simulation box. (**B**) End of the simulation after 30 ns total simulation time. The vitamin E molecule is totally dispersed within the PLA chains, from the surface to the core of the particle. The scale factor is the same as in (**A**) for better observation of the volume contraction. The whole simulation cell (280 × 280 × 280 Å) is displayed in A and B. (**C**) Zoomed-in cross-section of the particle. The water molecules are totally excluded from the hydrophobic core formed by the intricated PLA chains and the vitamin E molecules.

**Table 1 nanomaterials-10-02209-t001:** DPD repulsion parameters used for all the simulations involving water, PLA, Pam_3_CSK_4_ and Pam_1_CSK_4_ molecules.

	G	W	F	C	S	K	LA
G	78						
W	99.4	78					
F	26.1	91.3	78				
C	26.0	82.6	28.5	78			
S	28.9	61.4	31.8	26.1	78		
K	161.3	26.7	157.3	134.6	101.3	78	
LA	25.4	91.3	25.1	27.2	30.3	154.3	78

**Table 2 nanomaterials-10-02209-t002:** DPD repulsion parameters used for all the simulations involving vitamin E, water and PLA molecules.

	W	CHR	C5	LA
W	78			
CHR	169.6	78		
C5	115.0	25.9	78	
LA	91.3	25.0	25.3	78

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
