# Peer review of "Molecular Dynamics Studies of Poly(Lactic Acid) Nanoparticles and Their Interactions with Vitamin E and TLR Agonists Pam1CSK4 and Pam3CSK4"

_nanomaterials, 2020, doi:10.3390/nano10112209_

Round 1

Reviewer 1 Report

The paper by Megy and coworkers is devoted to the DPD simulation of the process of self-assembly of PLA nanoparticles and their incorporation of different kind of molecules exhibiting diverse molecular properties. Coarse graining and the use of DPD technique allow to describe large systems and to simulate time requiring processes that it would not be possible to simulate with all atom techniques. The most interesting aspect of the paper is the careful comparison of the results obtained from the simulations with the experimental test performed by the authors and with data available in literature.  

In general, the authors show that their simulation succeed in reproducing well experimental data. Moreover, they underline the inherent difficulties in simulating this kind of system and deeply discuss the problems in comparing experimental data with simulation which may have a different size and time scale. This discussion is in my opinion the most valuable part of the paper. 

I recommend the acceptation of this paper as is. 

Author Response

In response to Reviewer 1, some minor spell check was performed, and some typos were corrected (lines 53, 85, 131, 278, 289, 294, 304, 347, 358, 359, 390, 461, 474, 493, 497, 499, 519, 542, 547, 567, and 589). The (Biovia) mention for the Material and studio Software is now cited only once line 164 and has been removed from line 228.

As Reviewer 1 mentioned that the description of the methods could be improved, we mentioned the use of the Mesocite Module within Material Studio in the Materials and Methods section lines 203, 213, and 220. We also added the mention of water molecules in the legend of Figure 3 (line 308) and Figure 6 (line 406).

Reviewer 2 Report

This manuscript looks at the interactions of polylactic acid, a water insoluble polymer, with small amphiphilic peptides (Pam1CSK4 & Pam3CSK4) and hydrophobic vitamin E, using a combination of computational methods and experimental investigations, for applications in drug delivery. For most purposes, it  is an extension of the work reported by the authors in 2019, and the novelty is lost in the process, as the authors themselves admit it “Here we present an updated version of our molecular modeling results, with a more comprehensive approach including new molecules such as Pam1CSK4 and vitamin E, improved simulation parameters, longer simulation times, larger simulation systems and high-quality videos. We used some of our previous results to develop and validate the models presented in this study.” Pam3CSK4 encapsulation has been validated before and the current study includes vitamin E, which is the new and may be interesting component of this work. Thus, it is a stretch to claim that “This is, to our knowledge, the first study which investigates the behavior of PLA NPs using DPD calculations”, by calling their earlier investigations as, “We also presented early results of molecular modeling for mechanisms….”. I can recommend publication of this work with the following revisions:

The authors should remove statements such as the first study, as it is an update from their earlier work, as stated clearly in the manuscript. It does not lower the quality of this work, and their results do not require selling using fancy and misleading terminology.

The results of the DPD study are interesting and in line with what one would expect from interactions of a water insoluble polymer with amphiphilic molecules with different hydrophilic fractions and hydrophobic small molecules. The authors should highlight the significance or the influence of the confinement of water molecules in periodic boxes on the outcome of the self-assembly. The discussion does provide some light in this regard, but it will be good to elaborate it in light of the encapsulation and release behavior, as intended in initial setting of the goals.

It is very clear from the studies that the limitations imposed create very little and random interaction between PLA and peptides/vitamin E. One essential control missing is the completely hydrophilic molecule or small chain macromolecule to validate the extreme end of this case study from amphiphilic to hydrophobic. The results are easily predictable, but will clearly validate their interesting findings in this manuscript.

Author Response

In response to Reviewer 2, we totally agree and the sentence “This is, to our knowledge, the first study which investigates the behavior of PLA NPs using DPD calculations.” has been removed in totality.

For the influence of the confinement of water molecules in periodic boxes, as explained in the discussion, the number of water molecules is highly underestimated, resulting in a much more concentrated medium than the reality. This is one limitation of the DPD simulation method, but this is also the case in most full atomistic simulations. We agree that limiting the number of water molecules and thus, working in a much more concentrated medium, enhances the attraction/repulsion behavior of the different molecules present in the simulation. This leads to quite a segregationist model where the hydrophobic/hydrophilic interactions are somehow overestimated, and this might enhance the encapsulation behavior, which happens extremely fast in our simulations. This clearly is one of the limitations of our model, the other one obviously being the simulation of groups of atoms instead of individual atoms. Anyway, despite its inherent limitations, our simulation model seems robust enough to correctly describe the behavior of molecules with very different properties, as illustrated in our article.

For the release behavior, we are aware that our model is not really suited to describe such mechanisms. DPD simulations can describe molecular evolutions in the timescale of ms or at best ms (such as the encapsulation process), but release is a totally different process which occurs within hours or days. We actually have experimental data on the release of several small molecules encapsulated in PLA NPs. Release of Vitamin E in hydrophobic phases takes up to 2 weeks, and release of small antibiotic molecules such as Clindamycin and Rifampicine takes up to 48 hours. These timescales are clearly not reachable using molecular modeling simulation methods.

Another option would be to study the diffusion of an encapsulated molecule within a PLA NP. DPD is not really suitable for such studies, but this kind of simulation can be achieved using all atomistic models and can provide interesting data in regards of the release process.

These points have been added in the discussion section, lines 477 to 488.

Finally, in response to the study of completely hydrophilic molecules, we have already done some experimental work in this direction, for instance with small hydrophilic molecules such as CpG ODNs and Clindamycin. CPG ODNs are commercially available short synthetic single-stranded DNA molecules containing unmethylated CpG dinucleotides in particular sequence contexts and are very hydrophilic. Clindamycin is a small antibiotic molecule with mostly hydrophilic properties. CPG ODNs are extremely complicated to formulate with PLA NPs, and we have had no success in experimentally encapsulating them so far. Clindamycin has slightly more favorable properties, but so far, the encapsulation efficiency has been very low, and most of the molecules remain in the surrounding water. Of course, we highly suspect that the few encapsulated Clindamycin will also end up located at the surface of the PLA NPs, and not be buried inside.

A DPD study of the interaction of PLA NPs with such molecules would be a very good follow up to our work and will probably be started in a near future. Thus, we completely agree that this would validate the extreme end of this case study from amphiphilic to hydrophobic molecules. This will most certainly be developed in a next publication. These explanations have been included in the conclusion of the study, lines 570 to 578.
